behaviour/evolution

compression, Menzerath's Law, Zipf's Law of abbreviation, unsupervised clustering, *Hylobates*

**Author for correspondence:**
Dena J. Clink
e-mail: dena.clink@cornell.edu

# Brevity is not a universal in animal communication: evidence for compression depends on the unit of analysis in small ape vocalizations

## Dena J. Clink[1], Abdul Hamid Ahmad[2] and Holger Klinck[1]

[1]Center for Conservation Bioacoustics, Cornell Laboratory of Ornithology, Cornell University, Ithaca, NY, USA
[2]Faculty of Sustainable Agriculture, Universiti Malaysia Sabah, Sandakan Campus, Sabah, Malaysia

DJC, 0000-0003-0363-5581

Evidence for compression, or minimization of code length, has been found across biological systems from genomes to human language and music. Two linguistic laws—Menzerath's Law (which states that longer sequences consist of shorter constituents) and Zipf's Law of abbreviation (a negative relationship between signal length and frequency of use)—are predictions of compression. It has been proposed that compression is a universal in animal communication, but there have been mixed results, particularly in reference to Zipf's Law of abbreviation. Like songbirds, male gibbons (*Hylobates muelleri*) engage in long solo bouts with unique combinations of notes which combine into phrases. We found strong support for Menzerath's Law as the longer a phrase, the shorter the notes. To identify phrase types, we used state-of-the-art affinity propagation clustering, and were able to predict phrase types using support vector machines with a mean accuracy of 74%. Based on unsupervised phrase type classification, we did not find support for Zipf's Law of abbreviation. Our results indicate that adherence to linguistic laws in male gibbon solos depends on the unit of analysis. We conclude that principles of compression are applicable outside of human language, but may act differently across levels of organization in biological systems.

# 1. Introduction

Identifying universal principles underlying behaviour is a fundamental goal of evolutionary biology, and the identification of shared principles across diverse taxa can provide insights into the evolutionary mechanisms and physical constraints that shape behavioural diversity [1]. Human language has been considered unique among communication systems as it contains semantics, wherein sounds are combined in unique ways to confer meaning [2], but there are cases wherein animal vocalizations have been linked to particular behavioural contexts [3] and semantics in non-human communication remains a topic of debate [4]. Understanding the origins of human languages has intrigued scientists and philosophers for centuries, and a particularly useful approach for understanding the evolution of language has been the comparison of human and animal communication systems [5]. For example, many birds have the capacity for vocal learning—unlike the majority of non-human primates—and similar to human language birdsong is hierarchically structured [6]. Both birds and non-human primates have the capacity for producing calls with specific meaning, and the ability to combine multiple calls into a sequence which exhibits compositional syntax (wherein the order of the calls changes the meaning of the sequence [7]), but it has been proposed that the 'complexity and expressive power' of language is uniquely human [8].

Identifying 'core features' or universals of vocal communication across taxa is important for understanding the evolution of language [7]. One principle that has been proposed to be not only a universal in language, but across all aspects of animal behaviour [9], is that of compression. In information theory, the principle of compression predicts the minimization of length of code, and assigns strings as short as possible to represent the maximum amount of information [10]. Human language tends to follow statistical laws which can be explained by the principles of compression [11] and it has been suggested that compression in behavioural systems is the result of selection for energetic efficiency [9,12].

Exploring the universality of linguistic laws in animal models can help improve our understanding of the constraints on vocal production, and may provide insight into the processes that shape diversity in communication [13]. Such comparative approaches have been used to investigate Zipf's Law (not to be confused with Zipf's Law of abbreviation) wherein the frequency of a given word is proportional to its rank (meaning that the most commonly used word will occur approximately twice as often as the next most commonly used word [14]). Applications of Zipf's Law to animal communication systems have been applied with the overarching goal of finding 'indicators of potential structure' [15]. Communication systems in diverse taxa and modalities including dolphin (*Tursiops* spp.) whistles [16,17] and gorilla (*Gorilla gorilla*) gestures have been shown to conform to Zipf's Law [18]. However, even within closely related taxa, there were differences in conformity to Zipf's Law, as black-capped chickadee (*Parus atricapillus*) calls did not conform to Zipf's Law [19], whereas Carolina chickadee (*Poecile carolinens*) calls did, and the authors interpreted this as evidence that Carolina chickadee calls are more complex, which may be linked to differences in social complexity between the two species [20]. Two other linguistic laws, Menzerath's Law and Zipf's Law of abbreviation, have also received a substantial amount of attention in non-human animal models, and all three of these linguistic laws have been linked to compression [21].

Menzerath's Law is a linguistic law that states 'the greater the whole, the smaller its constituents' [22,23]. In human speech, the longer a word, the shorter the syllables that make up that word [22]. There is evidence for Menzerath's Law across diverse biological systems. For example, longer genomes tend to be made up of smaller chromosomes [24], genes which have a higher number of exons have exons of shorter average size [25], and the longer a protein, the smaller its structural domains [26]. Adherence to Menzerath's Law has been found in gelada (*Theropithecus gelada*) vocal sequences [12], wild chimpanzee (*Pan troglodytes*) hoots [27], chimpanzee gestures [28] and recently in vocal sequences of two species of crested gibbons (*Nomascus* spp. [29]). A better understanding of the way in which animals combine singular sounds into complex sequences, and the trade-offs that constrain vocal production [27], has important implications for understanding variation in acoustic complexity across taxa [12,30], and tests of Menzerath's Law in diverse taxa can provide insight into these processes.

Zipf's Law of abbreviation posits that more frequent elements in a communication system are shorter, or in human language that more frequently used words tend to be shorter [14,31]. The law of abbreviation provides evidence for compression, in that by employing shorter codes for more frequent words, more information can be transmitted more efficiently [32]. Zipf's Law of abbreviation has been found in most human languages examined [11]. In non-human animals, an inverse relationship between signal length and frequency of use may also be expected, if longer signals are costly to produce and the more frequent use of short signals maximizes coding efficiency [33]. One of the first documented tests of Zipf's Law of abbreviation in a non-human animal was in black-capped chickadees, and in this case, adherence to the law was dependent on the level of analysis: use of

different call types was not negatively correlated with the number of notes, but shorter bouts (which are composed of multiple calls with at least a 30 s break in between subsequent calls) were more frequent, meaning that bouts followed Zipf's Law of abbreviation [19].

Zipf's Law of abbreviation was subsequently shown in the repertoire of dolphin (*Tursiops* spp.) surface behaviours [34], vocalizations of Formosan macaques (*Macaca cyclopsis*) [33], the short-range vocalizations of four bat species [35], in subsets of the gestural repertoire of chimpanzees [28] and in penguin (*Spheniscus demersus*) vocalizations [36]. In crested gibbons, the most commonly used notes were the shortest, indicating that the use of distinct note types follows Zipf's Law of abbreviation [29]. But, Zipf's Law of abbreviation was not found in the vocal repertoire of two new world monkeys: common marmosets (*Callithrix jacchus*) and golden-backed uakaris (*Cacajao melanocephalus*) [37], ravens (*Carvus corax*) [38] or the full body gestures of chimpanzees [28] which may be related to the small repertoire sizes in these animals, or differences in function or context of use [37]. A recent study on rock hyraxes (*Procavia capensis*) found support for Zipf's Law of abbreviation in male but not female vocal repertoires, and also found a negative relationship between call amplitude and call usage, and the authors propose that for long-distance communication, costs of call amplitude may be more important than call duration [39].

Singing has independently evolved in non-human primates at least four times, as it is seen in indris (Indriidae), tarsiers (Tarsiidae), titi monkeys (Callicebinae) and gibbons (Hylobatidae [40]). The gibbons are the only singing ape, besides humans, which makes them a particularly interesting model for understanding the evolutionary roots of music and language [41], and unsurprisingly a substantial amount of work has been done investigating variation in gibbon calls [41–48]. Male gibbons engage in elaborate solo singing bouts that can last over an hour and are composed of a large repertoire of phrases comprising a discrete number of note types [49,50]. In all gibbon species examined, there appears to be a strong individual signature in the type and sequence of notes produced [41,49,51], and there is evidence that gibbons follow syntactical rules, as opposed to emitting notes at random [41]. In lar gibbons (*Hylobates lar*), notes were organized differently depending on whether calls were emitted under regular circumstances or when exposed to a predator model [52]. In the case of early morning solo vocalizations emitted by males, the basic units or phrases do not appear to bear any specific meaning [53], despite being highly complex [54], which provides an ideal system to investigate limits on vocal production in different call types emitted under presumably the same context. It may be that similar to gelada vocal sequences and birdsong, that the increasing complexity of male solos provides a more efficient way to deliver the same message [12,55], as opposed to human speech wherein combining different elements brings about different meaning.

A substantial amount of work has been done on quantifying and understanding variation in gibbon vocalizations, either focusing on adherence to syntactical rules in the notes of male solos [41,49,51,52], evidence for vocal individuality in male and female calls [42,46,56–60], trade-offs in the production of calls [61,62] or vocal flexibility [63,64], but to our knowledge, there has been little work investigating whether gibbon vocalizations adhere to linguistic laws. One of the first such tests was done on crested gibbons (*Nomascus* spp.), and the authors found support for both Zipf's Law of abbreviation and Menzerath's Law in note usage and organization [29]. Here, we provide a test of two linguistic laws in the solos of male Bornean gibbons (*Hylobates muelleri*). First, we test for evidence that gibbon phrases adhere to Menzerath's Law, and we predicted that the longer a particular phrase (in terms of the number of notes), the shorter the individual notes would be within the phrase. Second, we test for Zipf's Law of abbreviation in different phrases of the male solos.

For our test of Zipf's Law of abbreviation, we focused our analysis on the level of phrases of male gibbon solos, which comprise multiple notes. Notes in gibbon solos and syllables in human language are 'recombinable units' [19] which can be combined in almost infinite ways to produce phrases and words. Therefore, the structure of phrases in male solos is more analogous to words in human language than individual notes, and a phrase-level analysis may be more appropriate for exploring Zipf's Law of abbreviation. To identify unique phrase types within each male solo, we relied on an unsupervised clustering technique (affinity propagation clustering [65]) that is commonly used in genetics applications [66,67] and linguistics studies [68], and has recently been applied to acoustic data [69]. We predicted that, in accordance with Zipf's Law of abbreviation, there would be a negative relationship between phrase duration and frequency of use of that phrase type. Lastly, we investigated the degree of inter-individual variation in phrase types, given previously documented individual signatures in note type use [41,49,51] and phrases [56,70]. Our continued exploration of the applicability of statistical laws developed for human language in non-human systems—particularly in closely related species such as gibbons—can provide insights regarding the evolutionary history of universal linguistic patterns and the evolutionary precursors that led to the diversity of human languages [2].

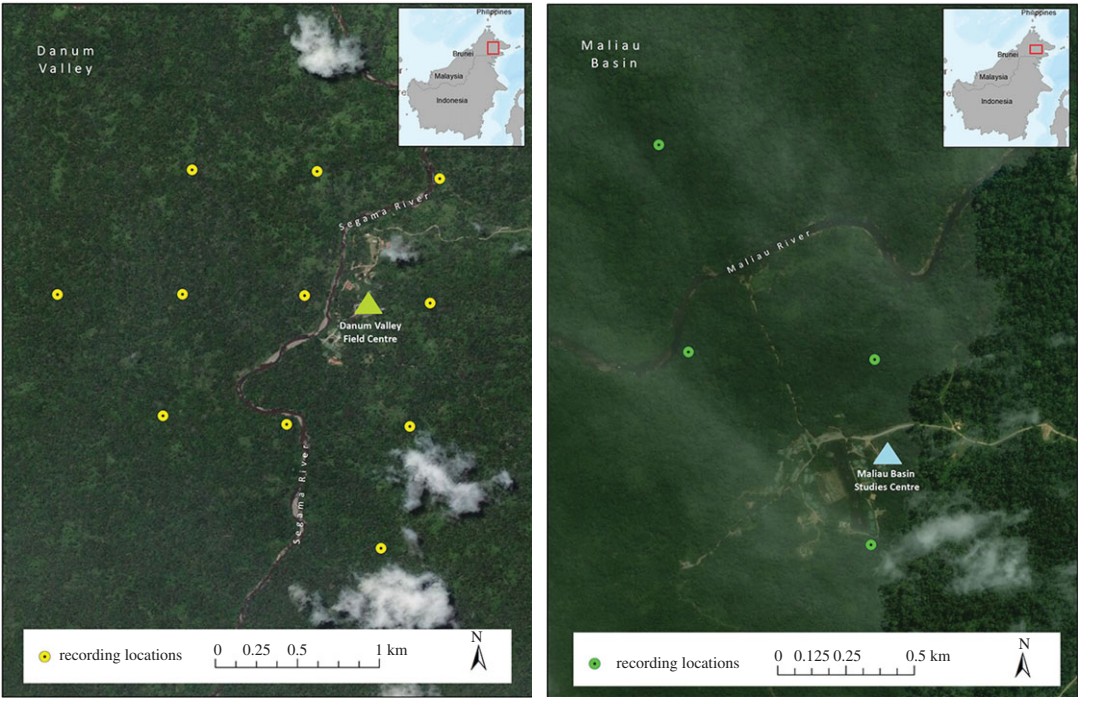

**Figure 1.** Map of autonomous recorder locations in Danum Valley Conservation Area and Maliau Basin Conservation Area, Sabah, Malaysia. Each point represents the location of a single Swift recording unit. Map was made using ArcGIS (ESRI) v. 10.5.1 (www.esri.com).

## 2. Methods

### 2.1. Study subjects

Mueller's Bornean gibbons (*H. muelleri*; hereafter gibbons) are pair-living hominoids found on the island of Borneo [40]. Both mated and unmated males engage in early morning solos [71] typically between the hours of 5.00–7.00 local time [72], with the solo performances of mated and paired males being indistinguishable to the human observer [71]. The mean duration for male solo bouts in Danum Valley Conservation Area, Sabah, Malaysia, was 41.4 min (range 6.5–88.1 min; [72]). In male gibbon solos, the smallest possible unit is termed a 'note' [73], and we defined 'phrases' as sequences of notes that are separated by breaks that are 2 s in duration or more, following Inoue *et al*. [41]. To date, the majority of analyses of gibbon solos have focused on quantifying patterns in various note types [29,41,49,51], but a few have done phrase-level analyses [52,64].

### 2.2. Data collection

Data were collected using Swift autonomous recording units [74] in Danum Valley Conservation Area (11 recording units; March–July 2018) and in Maliau Basin Conservation Area (four recording units; August 2019), Sabah, Malaysia (figure 1). The Swift units in Danum Valley Conservation Area recorded at a sampling rate of 16 kHz and the units in Maliau Basin recorded at a sampling rate of 48 kHz. For both recording locations, we recorded at a sample size of 16 bits and at a gain of 40 dB. The recording units were attached to a tree approximately 2 m above ground and collected data continuously. Recorders in Danum Valley Conservation Area were placed on a 750 m grid, and recorders in Maliau Basin Conservation Area were placed in presumed territories of different gibbon groups (greater than 600 m spacing). Early field tests indicate that detection distance of calling gibbons is approximately 400 m [75] using our recording settings. We considered high-quality recordings taken from each recorder as a different male based on the territorial nature of gibbons, the documented territory size in other gibbon species [76–78] and estimated detection distance of our recorders [75]; see Clink *et al*. [72] for more details.

## 2.3. Acoustic data processing

We first created long-term spectral average plots using the Matlab™-based program TRITON [79] to identify high-quality male solos (see [72] for details) from the long-term recordings. To avoid potential issues with pseudo-replication, we used the highest quality recording of a male solo from each of our recorders. In cases where there were multiple high-quality recordings on different days, we randomly chose a single one. See figure 2 for representative spectrograms of 40 s excerpts of solos from four males. Once we identified the location of male solos in our long-term data, we created high-resolution spectrograms in RAVEN PRO 1.6 [80], and identified each note using RAVEN PRO selection tables (figure 3). For the recordings with 16 kHz sampling rate, we made spectrograms with a 1024-point (64.0 ms) Hann window (3 dB bandwidth = 22.5 Hz), with 50% overlap, and a 1024-point discrete Fourier transform, yielding time and frequency measurement precision of 32 ms and 15.6 Hz. To obtain similar time and frequency resolution for the 48 kHz recordings, we downsampled the recordings to a 16 kHz sampling rate using ADOBE AUDITION 2020 before creating spectrograms.

For each note, we estimated start and stop time, along with the following features using the robust features in RAVEN PRO (which are more robust than traditional features to inter- and intra-observer reliability in feature selection [81]): 90% bandwidth, 90% duration and the lower and upper frequency bounds of the band containing 90% of the energy (minimum and maximum frequency). In line with previous researchers, we designated the break points between subsequent phrases as periods of silence of a duration of 2 s or longer (figure 3) [41]. All subsequent analyses were done in the R-programming environment version 3.6.2 [82].

## 2.4. Test for Menzerath's Law

To test for evidence for Menzerath's Law, we created a series of three generalized linear models with the mean note duration as the outcome using the 'lme4' package [83]. We first log-transformed both the mean note duration and number of notes in a phrase. The first model included only a random intercept for male (which we considered the null model), the second model included the number of notes in the phrase as a predictor along with a random intercept for male and the third model included the number of notes in the phrase along with a random intercept and random slope for each male. We compared models using Akaike information criterion adjusted for small sample sizes (AICc) and implemented in the R package 'bbmle' [84]. To test how well our top model fitted the data, we calculated a pseudo-$R^2$ value using the 'MuMIn' package [85].

## 2.5. Test for Zipf's Law of abbreviation

As outlined above, we distinguished between sequential phrases by identifying breaks that were 2 s in duration or longer, and then used two distinct feature extraction approaches for phrase classification. First, using the features estimated using the RAVEN PRO selection tables, we calculated the following for each of the Bornean gibbon phrases: number of notes, phrase duration, the mean, minimum and maximum frequency of all notes in the phrase, the lowest minimum frequency in the phrase, the highest maximum frequency in the phrase, the minimum, mean and maximum note bandwidth, and the minimum, mean and maximum note duration, resulting in a feature vector of length 13 for each phrase. For our second feature extraction method, we estimated Mel-frequency cepstral coefficients (MFCCs) for each of the phrases using the R package 'tuneR' [86] and the following settings: frequency range 0.5–1.2 kHz, window size = 0.25 s, number of cepstra = 12. We downsampled so that MFCCs were calculated for all recordings at a 16 kHz sampling rate. We then calculated the mean and standard deviation for each of the 11 cepstra over the entire phrase (omitting the first cepstrum as this is generally related to loudness of the signal [87] and not relevant for the present study), and added the duration of the phrase, which resulted in a feature vector of length 23 for each phrase.

To distinguish between putative phrase types, we used affinity propagation clustering [65], which is a commonly used unsupervised clustering method in genetics [66,67] and linguistics studies [68], and has recently been applied to acoustic data [69]. We implemented affinity propagation clustering using the R package 'APCluster' [88]. As we were interested in different phrase types within a single male, and given previously documented individual preference for different note combinations, we ran the unsupervised cluster analysis for each male solo individually. The number of clusters returned by affinity clustering can be influenced by the input preferences, so we systematically varied the input preferences using the '*q*'

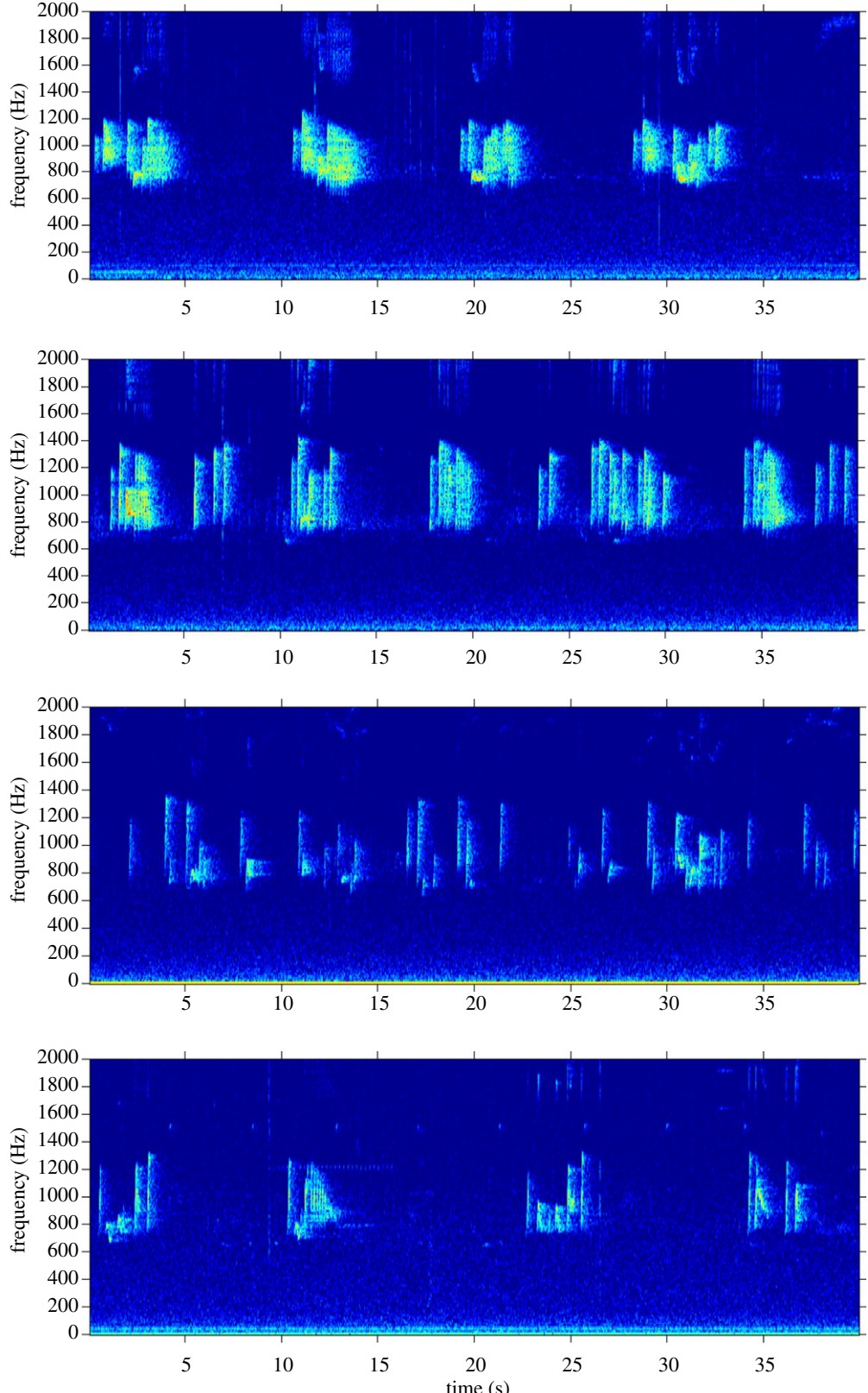

**Figure 2.** Representative spectrograms of 40 s excerpts of solos from four different Bornean gibbon males. Spectrograms were made using TRITON [79] with a 1600 point (100 ms) Hann window with 85% overlap. Background noise and harmonics were not removed.

input from 0 to 1 (in increments of 0.1), returned the cluster solutions and calculated a silhouette value for each using the 'silhouette' function from the R package 'cluster' [89]. We chose the '$q$' value which returned the highest silhouette value, a method known as adaptive affinity propagation clustering [90].

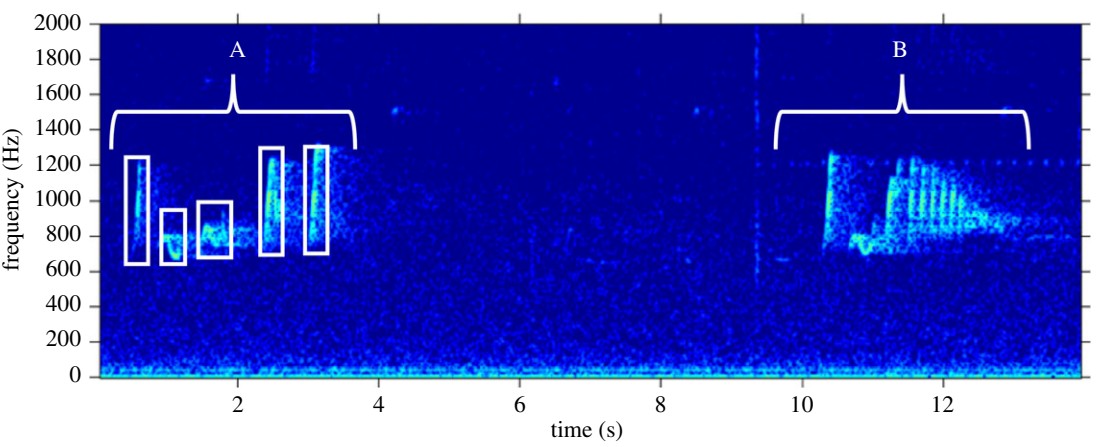

**Figure 3.** Representative spectrogram of phrases from a male Bornean gibbon solo. The white boxes indicate how notes were identified in the spectrogram, and the brackets and letters indicate distinct phrase types. Spectrograms were made using Triton [79] with a 1600 point (100 ms) Hann window with 85% overlap. This spectrogram is of the same solo that is shown in the bottom of figure 2 but is zoomed in on the time axis. Background noise and harmonics were not removed.

We compared adaptive and non-adaptive clustering, and found that adaptive clustering had higher mean silhouette coefficients based on features extracted from the spectrogram (mean silhouette coefficient: 0.24 versus 0.31), but returned a smaller amount of clusters per male (mean number of clusters: 13.2 versus 6.3). It has been suggested that lack of adherence to Zipf's Law of abbreviation in other species was owing to small repertoire size [37,38] so here we report the results of the analysis on the non-adaptive clustering, but for both clustering approaches, our qualitative inference was the same. To test which of our feature extraction methods resulted in better clustering, we compared silhouette coefficients for each method, and used a support vector machine (SVM [91]) to examine how well we could classify the putative phrase types, as assigned by affinity propagation cluster. SVMs were implemented in the R package 'e1071' [92].

To test for adherence to Zipf's Law of abbreviation in male solo phrases, we created a series of three generalized linear models with two separate outcomes (mean phrase duration and mean number of notes per phrase) using the 'lme4' package [83]. We log-transformed mean phrase duration, mean number of notes and frequency of phrase use to ensure our data conformed to the assumptions of the linear models. For each outcome, the first model included only a random intercept for each male (which we considered the null model), the second model included frequency of phrase use as a predictor and a random intercept for each male, and the third model including frequency of use as a predictor variable along with a random intercept and random slope for each male. We compared models using AICc implemented in the R package 'bbmle' [84]. To test how well our top models fit the data, we calculated a pseudo-$R^2$ value using the 'MuMIn' package [85].

## 2.6. Investigating individual signatures

Our data were collected using autonomous recorders, which means that all data were taken in the absence of human observers. Therefore, our identification of individuals is based on recording location only. To test for individual signatures in male solos, we used SVMs implemented in the 'e1071' [92] on both sets of features (features estimated from the spectrogram and MFCCs) and calculated classification accuracy using leave-one-out cross-validation.

# 3. Results

## 3.1. Unsupervised cluster analysis

We report the analysis solos from 13 male Bornean gibbons which comprised 2363 phrases (54 to 493 phrases per male) and 17 392 notes. We identified putative phrase types (vocal units composed of two or more notes which are separated by other phrases by 2 s or more of silence) within male solos using affinity propagation clustering, and verified cluster solutions based on silhouette coefficients. We found that the number of phrase

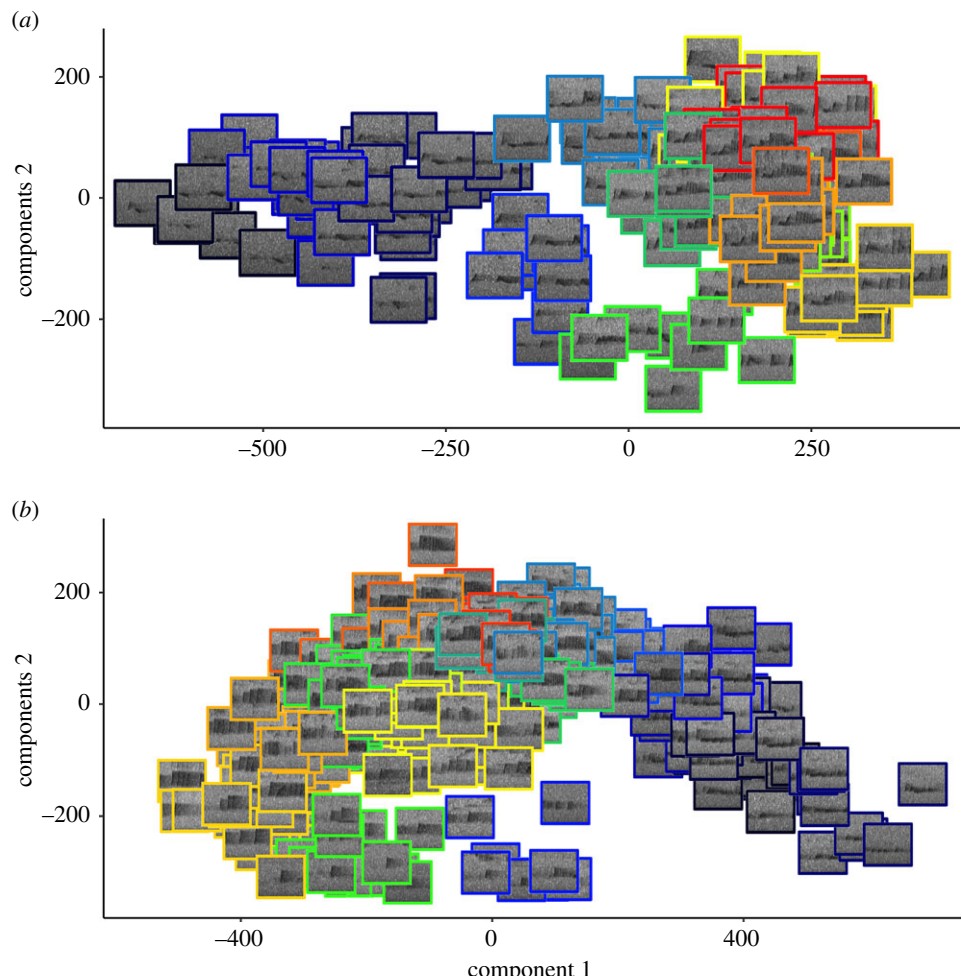

**Figure 4.** Visualization of the unsupervised clustering of male Bornean gibbon solos. Plots show the scatterplot of the first and second principal components of phrases from two male Bornean gibbon solos. A male solo from (*a*) Danum Valley and (*b*) Maliau Basin. Different coloured squares represent phrase type classification by affinity propagation clustering, and each box contains a spectrogram of a single phrase of the indicated phrase type. Spectrograms were made using the R package 'signal' [93].

types (clusters) per male solo varied from 8 to 26 (mean = 13.3 clusters), and the median phrase duration was 2.8 s. Cluster solutions based on features estimated from the spectrogram had higher silhouette coefficients (mean = 0.24, range = 0.17–0.35) than cluster solutions based on MFCCs (mean = 0.13, range = 0.09–0.19). We also found that we were able to classify putative phrase types with a higher accuracy using features from the spectrogram (mean classification accuracy of 73.6% versus 39.1% using SVM). Given these results, we used phrase type classifications based on features estimated from the spectrogram to test for Zipf's Law of abbreviation, but both methods of feature extraction led to a qualitatively similar result. We visually inspected phrase classifications using both spectrograms and principal component analysis biplots. See figure 4 for representative principal component biplot and representative spectrograms of phrases identified using affinity clustering. See figure 5 for the spectrograms of four phrases of the same phrase type from two different males, as determined using affinity propagation clustering. Exemplary phrase type classifications for each male in our dataset are available as the electronic supplementary material.

## 3.2. Menzerath's Law

We found strong support for Menzerath's Law in the phrases of male Bornean gibbon solos. Our top model included both number of notes in the phrase, along with a random intercept and slope for each male (estimate = −0.38, s.e. = 0.07; figure 6). Our top model performed substantially better than the null model, which only contained a random intercept and slope for each male (ΔAICc = 900.2, less than 0.001% model weight). We calculated a pseudo-$R^2$ value and found that the number of notes in a phrase explained a substantial amount of the variance (19%) and the entire model (predictor and

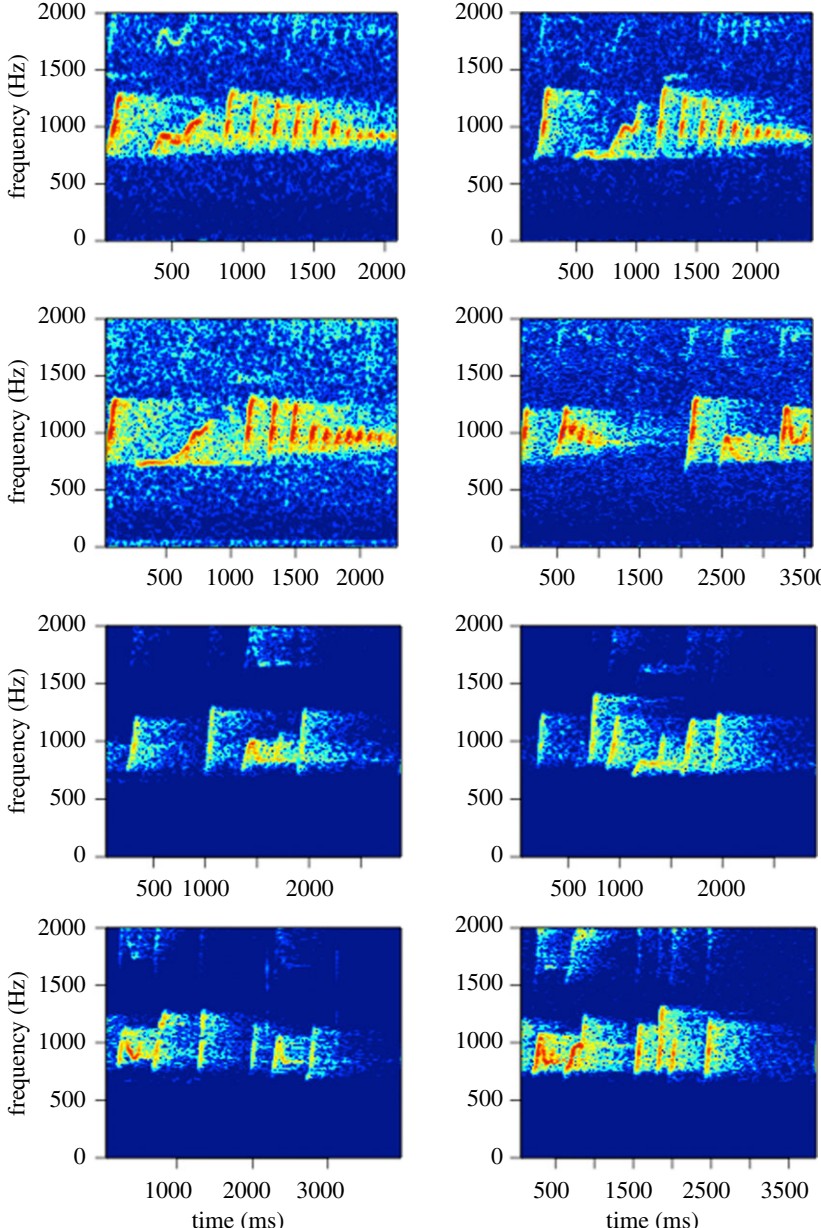

**Figure 5.** Representative spectrograms of four different phrases classified as the same phrase type from two different male Bornean gibbon solos (top four spectrograms are from a male at Danum Valley Conservation Area and the bottom four are from a male at Maliau Basin Conservation Area). Spectrograms were made with the 'phonTools' R package [94].

random effects) explained 67% of the variance. The high amount of variance explained by the random effects indicates that there was a substantial amount of variation among males.

## 3.3. Zipf's Law of abbreviation

To test for Zipf's Law of abbreviation, we first created a series of generalized linear models including mean phrase duration as the outcome variable to test for Zipf's Law of abbreviation. Using information criterion and AIC model selection, we found that our top model for the mean phrase duration included both the predictor variable frequency of use along with random intercept and slope (estimate = 0.39, s.e. = 0.09; figure 7). The top model comprised 99% of the model weight and performed substantially better than the null model ($\Delta$AICc = 51.8; less than 1% of model weight). We calculated a pseudo-$R^2$ value and found that the frequency of use explained a substantial amount of the variance (21%) and the entire model (predictors and random effects) explained 49% of the variance.

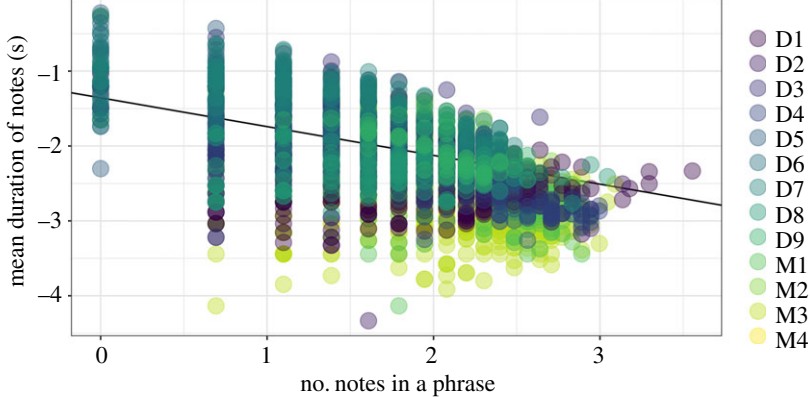

**Figure 6.** Support for Menzerath's Law in male Bornean gibbon solos phrases. The plot shows the relationship between the number of notes in a phrase, and the mean duration of the notes. Both x- and y-axes are log-transformed, and the different coloured points represent different males. The D and M in the legend denotes whether the male was recorded at Danum Valley or Maliau Basin Conservation Area. The black line represents the regression line from the top model chosen based on AIC comparison.

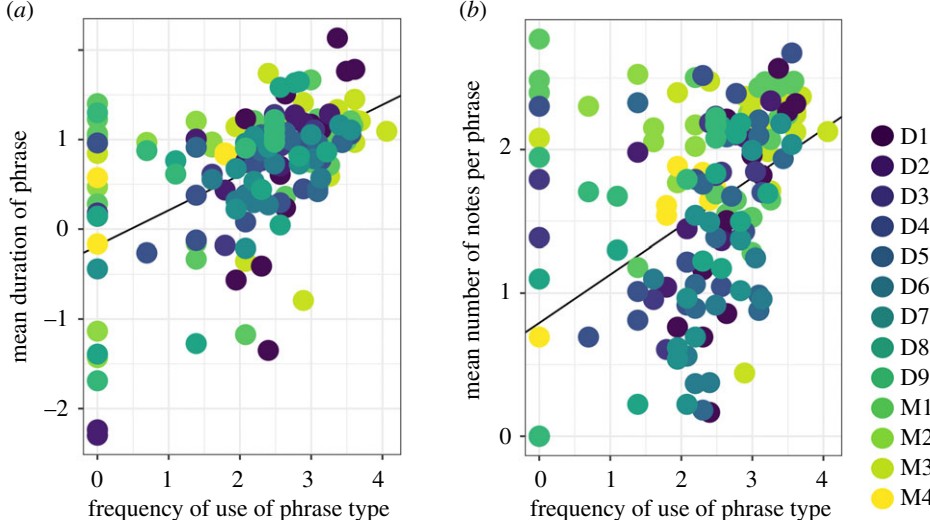

**Figure 7.** Lack of support for Zipf's Law of abbreviation in phrases of male gibbon solos. (*a*) Relationship between the mean phrase duration (s) and frequency of use of a particular phrase type and (*b*) relationship between the mean number of notes per phrase and frequency of use. Both axes are displayed on a log-scale, and different coloured points indicate phrases from different males. The D and M in the legend denote whether the male was recorded at Danum Valley or Maliau Basin Conservation Area. The black line represents the regression line from the top model based on AIC comparison. See text for details.

We also created a series of linear models with the mean number of notes per phrase as the outcome. Using AIC model comparison, we found that the top model comprised 94% of the model weight and included frequency of phrase type as a predictor (estimate = 0.34, s.e. = 0.07; figure 7) along with a random slope and intercept for each male. The top model performed substantially better than the null model ($\Delta$AICc = 27.1; less than 1% of model weight). The pseudo-$R^2$ value indicated that the predictor explained 21% of the variance and the entire model (predictors and random effects) explained 47% of the variance.

## 3.4. Individual signatures

To investigate the potential for individual signatures in male solo phrases, we trained an SVM to classify phrases to individuals and validated using a leave-one-out cross-validation approach. We found that we were able to classify phrases to the correct male with a 79.0% accuracy features estimated from the spectrogram and an 84.8% accuracy using MFCCs; both approaches resulted in classification accuracy that was substantially higher than by chance (1 out of 13 males = 8%).

# 4. Discussion

We found that the phrases within male gibbon solos followed Menzerath's Law, as there was a strong negative correlation between the number of notes in a phrase and the mean duration of the notes, or in other words for gibbon solo phrases 'the greater the whole, the smaller its constituents' [23]. We did not find support for Zipf's Law of abbreviation as there was not a negative relationship between duration of a signal and the frequency of use. In fact, we found a pattern opposite to that predicted by the Law, as phrase types that were longer (or had more notes) tended to be used more often. In agreement with previous studies on vocal individuality in gibbons [42,46,56–60], we found that males had strong individual signatures in their phrases. Our results indicate that compression may have shaped male gibbon vocal communication at one level—the construction of notes into phrases—but not at another level, in the pattern of use of different phrases within a solo. It is possible that compression is not universally applicable in vocal communication systems, or that other selection pressures were stronger in shaping phrase organization within male solos [9].

Our support for Menzerath's Law is in accordance with previous investigations regarding the trade-off between the mean note duration and phrase length in non-human primates [12,27]. In addition, other trade-offs in the vocal production of gibbons have been shown to occur. For example, in the trill of the Bornean gibbon female contribution to the duet, there is a trade-off between trill rate and bandwidth [61], and in lar gibbons, the maximum fundamental frequency decreased with an increase in signal duration [62]; this was also the case for certain phrases in chimpanzee vocal sequences [27]. Previous authors have suggested that these trade-offs are reflections of biomechanical constraints on vocal production [27,61,62]. Although we found support for Menzerath's Law, the underlying processes that led to the observed pattern are not clear. It is possible that the observed pattern is the result of selection for compression or coding efficiency (as outlined in [12,95]), or it could be the result of biomechanical constraints on the production of notes, and that it is difficult for male gibbons to produce long phrases with longer notes.

Zipf's Law of abbreviation is expressed through a negative relationship between the length of words (or acoustic signals) and how often they are used [31]. Across diverse taxa, there have been patterns consistent with this Law [28,33,35,36], patterns which are inconsistent with the Law [38] and in some cases adherence to the Law was dependent on the unit of analysis [19] or sex [39]. One of the first documented cases of a significant positive relationship between signal duration and frequency of use was in a subset of the chimpanzee gestural communication repertoire [28], and here, we show that phrases of male gibbon solos follow a pattern opposite to that predicted by Zipf's Law of abbreviation. A previous test of Zipf's Law of abbreviation in gibbons focused on individual note types [29] and found patterns consistent with Zipf's Law of abbreviation, whereas we focused on phrases which comprise multiple notes. In the foundational work by Hailman *et al.* [19] on chickadees, adherence to Zipf's Law of abbreviation was also dependent on the unit of analysis, but in this case, the smallest units (call types) did not reflect the pattern but larger units (bouts) did, which is opposite to what is seen in gibbons. Pressure for efficiency may vary across levels of organization and may also be species-specific. It appears that compression is important for notes in gibbon solos, but less so for the more complex phrases.

The evolutionary implications of adherence (or lack thereof) to Zipf's Law of abbreviation are unclear, and in gibbons, it is possible that selection pressure for phrases of longer duration have been stronger than the selection pressure for compression. Although the function of male solos remains a topic of debate [47,96,97], it is clear that they are a long-distance signal that transmits information regarding the caller to conspecifics, and most likely information about caller location and identity. It is possible that male solos provide honest information about caller fitness, as males with higher androgen levels have higher frequency calls [45] and older female gibbons have lower frequency calls [62]. In other taxa, the duration of signals has been shown to be an honest indicator of caller fitness. For example, in the grey tree frog (*Hyla versicolor*), the offspring of males that produced longer calls did better than the offspring of males with shorter calls [98]. In both chimpanzees [99] and geladas [100], the rate of calling rather than call duration were found to vary with caller quality. As Bezerra *et al.* [37] highlight, selection may favour efficiency in acoustic signals, but there are also other potentially opposing selective pressures that will result in patterns that are not consistent with that of compression, or as in the case of rock hyraxes other factors, such as call amplitude, may be more important determinants of call usage than call duration [39].

It has been proposed that patterns of compression may not be as prevalent in long-distance calls, particularly in noisy environments where selection can drive signals in the opposite direction [9]. For example, elongation of signals or increased redundancy in signals may combat the effects of environmental noise [101]. Therefore, it is possible that we did not find support for Zipf's Law of abbreviation because selective forces which oppose compression, and maximize transmission in noisy environments, were more important in shaping male solo phrase organization. As Demartsev *et al.* [39]

suggest, inconsistent support for Zipf's Law of abbreviation in animal communication systems may also be related to the fact that Zipf's Law of abbreviation uses 'word length' as the basic linguistic unit [35], but that animal vocal communication systems lack analogues to words, and animal call types are not truly analogous to words in human language. In addition, animals and human communicate in acoustic environments that are much different, with the majority of human speech adapted for short-distance communication, whereas many animal communication systems occur at variable transmission distances, and often across long-range distances [102]. Therefore, the selective pressures on human language and non-human animal communication are likely to be different.

Another possible explanation for our lack of support for Zipf's Law of abbreviation may be related to the graded nature of male solo phrases which made it difficult to detect patterns consistent with Zipf's Law of abbreviation. The average silhouette coefficient for our cluster analysis was 0.24, confirming that phrases of male solos are intergraded. But, our ability to discriminate between putative phrase types with approximately 74% accuracy using supervised classification provides evidence that phrases types can be effectively categorized. Our results are in agreement with other analyses of vocal mammalian repertoires that show call types tend to be intergraded [103,104], even in call types that were previously thought to be discrete [105]. Another potential complication is that mated and unmated male gibbons engage in solo bouts, although previous authors have noted that there are no notable differences between the solos of mated and unmated males [41,106]. As we relied on passive acoustic monitoring, we had no way to determine if the males we recorded were mated or unmated. It is possible that selection pressures act differently on solo bouts of mated versus unmated males, and further studies with known status and identity of male subjects might be informative.

Adherence of gibbon solos to Menzerath's Law is most likely related to morphological and physiological constraints on vocal production. The lack of adherence of gibbon solos to Zipf's Law of abbreviation may be the result of opposing selective forces for effective transmission of acoustic signals in a noisy tropical forest environment. Importantly, mathematical support for Menzerath's and Zipf's Law does not require assumptions about the linguistic or communicative value of the elements (i.e. whether individual signals convey any meaning), so the fact that individual phrases within male solos presumably do not convey different meanings does not preclude the application of these laws [12]. To understand whether gibbon solos are an exception to the universal rule of compression in behaviour [9], further work on different gibbon species, as well as on their short-range vocal repertoire, will be informative. In addition, further applications in a variety of primate and non-primate taxa will provide insights into the broad-scale applicability of linguistic laws outside of human language systems, and perhaps inform understanding of universal principals of vocal communication and behaviour more broadly.

Ethics. The research presented here adhered to all local and international laws. Institutional approval was provided by Cornell University (IACUC 2017-0098). Sabah Biodiversity Centre and the Danum Valley and Maliau Basin Management Committee provided permission to conduct the research.

Data accessibility. All R code and data needed to recreate analyses are available from the Dryad Digital Repository: https://dx.doi.org/10.5061/dryad.wstqjq2h8 [107]. Sound files are located in the persistent repository at the following URL: https://cornell.box.com/s/0pqxw6gxvg69nb1ipcy25j4zs5r0q2ua.

Authors' contributions. D.J.C., A.H.A. and H.K. developed the study design. D.J.C. did data collection and analysis. D.J.C., A.H.A. and H.K. contributed to writing of the manuscript, and all authors read and approved the final manuscript.

Competing interests. The authors declare that they have no competing interests.

Funding. Funding for this research was provided by a Fulbright ASEAN Research Award for US Scholars (no award number given).

Acknowledgements. We thank Yoel Majikil for assistance with data collection for this project, and Xabi Darthayette for his help annotating male solos. We gratefully acknowledge Russell A. Charif for his input on earlier versions of this manuscript and Ashakur Rahaman for his assistance with making the map. We also thank two anonymous reviewers for their helpful comments which substantially improved the manuscript.

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
