## [Reviewer comments · Royal Society Open Science]

Review History

RSOS-200151.R0 (Original submission)

Review form: Reviewer 1

Is the manuscript scientifically sound in its present form?

Yes

Are the interpretations and conclusions justified by the results?

Yes

Is the language acceptable?

Yes

Do you have any ethical concerns with this paper?

No

Have you any concerns about statistical analyses in this paper?

No

Recommendation?

Major revision is needed (please make suggestions in comments)

Comments to the Author(s)

This is a very important contribution to research on the law of abbreviation and Menzerath's law in other species. This research builds directly on findings of these laws at the level of note usage and organization in crested gibbons. The authors extend the analysis to Bornean gibbons regarding phrases (made of notes) as a "words" and find that Menzerath's law holds while the law of abbreviation does not; actually, they find the inverse of the law of abbreviation (a positive correlation between frequency of use and phrase duration). The article is making a very important contribution towards understanding the conditions under which the principle of compression manifests. The article is of high technical quality, well-written and thus deserves publication. However, I have some particular concerns that I explain below (I think they can be addressed easily).

1. THEORETICAL BACKGROUND

The current reasoning is somewhat superficial from a theoretical standpoint.

Abstract: "Zipf's law of abbreviation (which predicts a negative relationship between signal length and frequency of use). Indeed, Zipf's law of abbreviation does not predict in a theoretical sense. What really predicts is the principle of compression. The law of abbreviation is a manifestation of compression. Making predictions based on the law of abbreviation per se is just inductive reasoning.

(but in lines 54-55 the authors are right when stating that "the principle of compression predicts ...")

Abstract: "are linked to compression" -> "are predictions of compression".

"We did not find support for Zipf's law of abbreviation – which predicts a negative relationship between duration of a signal and the frequency of use." It would be better to state that "We did not find [evidence of compression for the] relationship between duration of a signal and the frequency of use."

Please, revise the use of "predict/ion..." everywhere.

The title "Brevity is not a universal in animal communication: lack of evidence for efficient coding in phrases of small ape vocalizations"

Gibbons may be coding efficiently, but not to maximize compression but to maximize transmission success, indicate fitness, ... (as reviewed by the authors in the discussion section).

A theoretically sounder title could be "Brevity is not a universal in animal communication: lack of evidence for [compression] in phrases of small ape vocalizations"

The problem of that title would be that Menzerath's law is still a prediction of compression (Gustison et al and <https://arxiv.org/pdf/1906.01545.pdf>).

Furthermore, the issue that the law of abbreviation is not universal in animal communication is not totally new. It failed in chimpanzees for body gestures, in female hyraxes (but no in males) and also failed in human heraldry.

Paradoxically, "brevity is not universal" (these gibbons do not show the law of abbreviation) but show another prediction of compression, i.e. Menzerath's law.

It is convenient to distinguish universality of the applicability of the principle of compression at two levels: (a) species level (b) all behaviors of a species (c) ...

I do not think that it is difficult to find violations of the predictions of compression in some dimension of the behavior of a species even in humans. What it will be really difficult is to show that no aspect of a species is driven by compression, even in experimental conditions or ecological contexts where compression should be favoured. That is the real point.

line 376-378 "It is possible that compression is not universally applicable in vocal communication systems, or that other selection pressures were stronger in shaping phrase organization within male solos."

A suitable citation here is Ferrer-i-Cancho et al in Cognitive Science (that is revised later on the article in detail).

line 386-388 "Although we found support for Menzerath's law, the underlying processes that led to the observed pattern are not clear. It is possible that the observed pattern is the result of selection for coding efficiency,"

Indeed, Gustison et al (cited) proposed a generalization of compression behind Menzerath's law and should be cited here.

The most updated version of the mathematical theory of the law abbreviation (and Menzerath's law) is found here

<https://arxiv.org/pdf/1906.01545.pdf>

Relevant, for instance, in line 458.

lines 91-95 "A recent study on rock hyraxes did not find support Zipf's law of abbreviation, but rather found a negative relationship between call amplitude and call usage, and the authors posit that for long-distance communication costs of call amplitude may be more important than call duration [29]."

The wording is somewhat confusing. First, that study found support for Zipf's law of abbreviation in males and the opposite trend in females. Second, a negative relationship between call amplitude and call usage is a form of law of abbreviation (there length or duration is replaced by a magnitude, amplitude in this case) and therefore is a prediction of compression based on the general mathematical theory in <https://arxiv.org/pdf/1906.01545.pdf>

References to research on hyraxes later on in the article need to be revised accordingly.

An important mathematical finding (<https://arxiv.org/pdf/1906.01545.pdf>) is that compression predicts that the correlation between frequency and magnitude cannot be (strictly positive) positive. Interestingly, optimal coding is possible even when the correlation is zero (non significant according to a two-sided correlation test). Then, although the positive correlation found in this article and previous research is not consistent with compression, compression cannot be excluded a priori for cases where no significant correlation is found (the two new world primates of Bezerra et al, ravens,...).

2. THE LAW OF ABBREVIATION

The authors investigate the relationship between phrase frequency and its duration arguing that phrases resemble human "words" more than notes.

For that reason, and for the sake of completeness and robustness, the authors should also investigate the relationship between phrase frequency and number of notes within the phrase, that is closer to the typical setup used for testing the law of abbreviation in human languages, where word length in letters or syllables is used. My prediction is that the analysis will confirm the current findings of an anti-law of abbreviation and thus the results will be more convincing for all sorts of language researchers.

3. REVIEW OF THE LITERATURE

The article lacks references to research on these laws in other species. For instance, concerning the law of abbreviation, the pioneering research by Hailman (about 30 years ago) on chick-a-dee's is missing. The recent findings by penguins (Favaro et al, Biolody letters) are also missing.

Such a problem can be fixed checking a comprehensive bibliography:

https://www.cs.upc.edu/~rferrericancho/laws_of_language_outside_human_language.html.

line 81 "One of the first documented reports of Zipf's law of abbreviation in a nonhuman animal was in the repertoire of dolphin surface behaviors" needs to be revised accordingly.

In addition, the article would benefit from a wider perspective, other comparative research based on linguistic laws that are not the two laws considered by the authors. That would help to not lose historical perspective. For instance, the work by McCowan et al through Zipf's law for word frequencies (see biblio above).

line 92-94: Other well-known exceptions to the law are discussed here (fully body gestures in chimpanzees, heraldry,...)

<https://arxiv.org/pdf/1906.01545.pdf>

The case of the chimpanzees is particularly relevant for this article. It is discussed later on in the discussion section but could be cited quickly here.

4. HUMAN LANGUAGE UNIQUENESS

The statement "Human language is unique among communication systems as it contains semantics, wherein sounds are combined in unique ways to confer meaning [2]." is false in the way in its current formulation.

A possible counterexample (given the authors' loose statement): meaning in dolphins whistle sequences

- Ferrer-i-Cancho, R. & McCowan, B. (2009). A law of word meaning in dolphin whistle types. *Entropy* 11 (4), 688-701.

- Ferrer-i-Cancho, R. & McCowan, B. (2012). The span of correlations in dolphin whistle sequences. *Journal of Statistical Mechanics*, P06002.

Furthermore, could the authors state clearly the rigorous statistical test (traditional p-value testing? information theoretic model selection?...) that Berwick et al have used to determine that human language is unique in this regard? (I would like to apply it to my own data). If such a test did not exist the statement would be a projection of anthropocentric culture, not real science.

p. 8 "the diversity of language that is uniquely human [17]." Reference by Heesen et al is not about the diversity of language or the uniqueness of human language. It is rather the opposite, a challenge to anthropocentric biases and an attempt to provide unifying approaches.

The same applies to "the 'complexity and expressive power' of language is uniquely human [5]." I would really like to have such a statistical procedure.

In addition, indicating the actual range of species in which the test has been applied to determine the uniqueness of human language would really help. Have dolphins, humpback whales, belugas, ..., bats, geladas, ... been considered to reach such a conclusion?

It is also important to clarify, for a given species the modality or kind of vocalization (for instance, dolphins whistles versus click trains) that has been used to support the uniqueness claim.

Review form: Reviewer 2

Is the manuscript scientifically sound in its present form?

Yes

Are the interpretations and conclusions justified by the results?

Yes

Is the language acceptable?

Yes

Do you have any ethical concerns with this paper?

No

Have you any concerns about statistical analyses in this paper?

No

Recommendation?

Accept with minor revision (please list in comments)

Comments to the Author(s)

This is an interesting paper investigating whether structural trade-offs known as Menzerath's and Zipf's laws apply to male gibbon songs. I believe that this study is a valuable addition to what we know about this topic. I only have few minor comments.

Abstract: Please provide the name of the study species (not just "gibbons").

L 35-48: I think that the opening paragraph, especially the second part of it that relates to human semantics, is not very relevant to what this paper is dealing with (e.g. this paper does not look at the relationship between call structure and meaning/function).

L 70-73: It is not clear how these compression-related constraints relate to syntax.

L 139-141: The last hypothesis does not seem to fit or follow the two previous ones related to trade-offs. I would suggest linking it to the previous two hypotheses better.

L 141-144: This is kind of repetition of what you said at the beginning of the Introduction.

Methods: Please start the Methods section with providing information on study subjects and study site, before describing data collection.

Figure 3: Please put full axis titles. Axis x title is missing.

L 308-309: While I understand that Spearman's rank correlation was used to be consistent with previous studies, this analysis is inappropriate due to, as you know, not dealing with the non-independence of datapoints, and therefore should not be reported (the fact that such analyses were provided by previous studies is not a justification for using them).

L 338: It is unclear to me what is meant by "coding efficiency" here, and what kind of selection pressures would act on it (ultimately leading to the described compression patterns). Maybe expand on this.

L 391-398. A substantial part of this is a repetition of what was said in the Introduction. Please try to make the text more concise by avoiding such redundancy.

L 399-409. It looks like the differences between the results of your and the other gibbon study are due to the differences in the call unit selected for analysis. But the title of the paper reads like your study shows a different pattern compared with that previous study. This does not need to be the case, since, as you say, you did a different kind of analysis. So the title framed the way you did might be a bit misleading.

L 409-421: As mentioned above, this part of the study seems to be disconnected from the rest. It is interesting but it reads like from a different paper. Maybe try to integrate this aspect better with the rest of the MS.

L 426-429: But maybe, as mentioned above, you would find this if you used a different unit for analysis.

L 437-438: Maybe just "different" rather than "much different". E.g. Menzerath's law seems to be supported by data from several animal taxa.

Decision letter (RSOS-200151.R0)

18-Feb-2020

Dear Professor Clink

On behalf of the Editors, I am pleased to inform you that your Manuscript RSOS-200151 entitled "Brevity is not a universal in animal communication: lack of evidence for efficient coding in phrases of small ape vocalizations" has been accepted for publication in Royal Society Open Science subject to minor revision in accordance with the referee suggestions. Please find the referees' comments at the end of this email.

The reviewers and handling editors have recommended publication, but also suggest some minor revisions to your manuscript. Therefore, I invite you to respond to the comments and revise your manuscript.

- Ethics statement

- Data accessibility

If you wish to submit your supporting data or code to Dryad (<http://datadryad.org/>), or modify your current submission to dryad, please use the following link:
<http://datadryad.org/submit?journalID=RSOS&manu=RSOS-200151>

- Competing interests

- Authors' contributions

- Acknowledgements

- Funding statement

Because the schedule for publication is very tight, it is a condition of publication that you submit the revised version of your manuscript before 27-Feb-2020. Please note that the revision deadline will expire at 00.00am on this date. If you do not think you will be able to meet this date please let me know immediately.

- 1) A text file of the manuscript (tex, txt, rtf, docx or doc), references, tables (including captions) and figure captions. Do not upload a PDF as your "Main Document";
- 2) A separate electronic file of each figure (EPS or print-quality PDF preferred (either format should be produced directly from original creation package), or original software format);
- 3) Included a 100 word media summary of your paper when requested at submission. Please ensure you have entered correct contact details (email, institution and telephone) in your user account;

- 4) Included the raw data to support the claims made in your paper. You can either include your data as electronic supplementary material or upload to a repository and include the relevant doi within your manuscript. Make sure it is clear in your data accessibility statement how the data can be accessed;
- 5) All supplementary materials accompanying an accepted article will be treated as in their final form. Note that the Royal Society will neither edit nor typeset supplementary material and it will be hosted as provided. Please ensure that the supplementary material includes the paper details where possible (authors, article title, journal name).

If your manuscript is newly submitted and subsequently accepted for publication, you will be asked to pay the article processing charge, unless you request a waiver and this is approved by Royal Society Publishing. You can find out more about the charges at <https://royalsocietypublishing.org/rsos/charges>. Should you have any queries, please contact opscience@royalsociety.org.

Kind regards,
Andrew Dunn
Royal Society Open Science Editorial Office
Royal Society Open Science
opscience@royalsociety.org

on behalf of Dr Claudia Wascher (Associate Editor) and Kevin Padian (Subject Editor)
opscience@royalsociety.org

Associate Editor Comments to Author (Dr Claudia Wascher):

Associate Editor: 1

Comments to the Author:

The presented research investigates abbreviation laws (Zipf's and Menzerath's law) in vocalisations of Bornean gibbons. Both reviewers find the research an important contribution to the field and well presented. The reviewers have some minor comments which should be addressed prior to publication.

Reviewer comments to Author:

Reviewer: 1

Comments to the Author(s)

This is a very important contribution to research on the law of abbreviation and Menzerath's law in other species. This research builds directly on findings of these laws at the level of note usage

and organization in crested gibbons. The authors extend the analysis to Bornean gibbons regarding phrases (made of notes) as a "words" and find that Menzerath's law holds while the law of abbreviation does not; actually, they find the inverse of the law of abbreviation (a positive correlation between frequency of use and phrase duration). The article is making a very important contribution towards understanding the conditions under which the principle of compression manifests. The article is of high technical quality, well-written and thus deserves publication. However, I have some particular concerns that I explain below (I think they can be addressed easily).

1. THEORETICAL BACKGROUND

The current reasoning is somewhat superficial from a theoretical standpoint.

Abstract: "Zipf's law of abbreviation (which predicts a negative relationship between signal length and frequency of use). Indeed, Zipf's law of abbreviation does not predict in a theoretical sense. What really predicts is the principle of compression. The law of abbreviation is a manifestation of compression. Making predictions based on the law of abbreviation per se is just inductive reasoning.

(but in lines 54-55 the authors are right when stating that "the principle of compression predicts ...")

Abstract: "are linked to compression" -> "are predictions of compression".

"We did not find support for Zipf's law of abbreviation – which predicts a negative relationship between duration of a signal and the frequency of use." It would be better to state that "We did not find [evidence of compression for the] relationship between duration of a signal and the frequency of use."

Please, revise the use of "predict/ion..." everywhere.

The title "Brevity is not a universal in animal communication: lack of evidence for efficient coding in phrases of small ape vocalizations"

Gibbons may be coding efficiently, but not to maximize compression but to maximize transmission success, indicate fitness, ... (as reviewed by the authors in the discussion section).

A theoretically sounder title could be "Brevity is not a universal in animal communication: lack of evidence for [compression] in phrases of small ape vocalizations"

The problem of that title would be that Menzerath's law is still a prediction of compression (Gustison et al and <https://arxiv.org/pdf/1906.01545.pdf>).

Furthermore, the issue that the law of abbreviation is not universal in animal communication is not totally new. It failed in chimpanzees for body gestures, in female hyraxes (but no in males) and also failed in human heraldry.

Paradoxically, "brevity is not universal" (these gibbons do not show the law of abbreviation) but show another prediction of compression, i.e. Menzerath's law.

It is convenient to distinguish universality of the applicability of the principle of compression at two levels: (a) species level (b) all behaviors of a species (c) ...

I do not think that it is difficult to find violations of the predictions of compression in some dimension of the behavior of a species even in humans. What it will be really difficult is to show that no aspect of a species is driven by compression, even in experimental conditions or ecological contexts where compression should be favoured. That is the real point.

line 376-378 "It is possible that compression is not universally applicable in vocal communication systems, or that other selection pressures were stronger in shaping phrase organization within male solos."

A suitable citation here is Ferrer-i-Cancho et al in Cognitive Science (that is revised later on the article in detail).

line 386-388 "Although we found support for Menzerath's law, the underlying processes that led

to the observed pattern are not clear. It is possible that the observed pattern is the result of selection for coding efficiency,"

Indeed, Gustison et al (cited) proposed a generalization of compression behind Menzerath's law and should be cited here.

The most updated version of the mathematical theory of the law abbreviation (and Menzerath's law) is found here

<https://arxiv.org/pdf/1906.01545.pdf>

Relevant, for instance, in line 458.

lines 91-95 "A recent study on rock hyraxes did not find support Zipf's law of abbreviation, but rather found a negative relationship between call amplitude and call usage, and the authors posit that for long-distance communication costs of call amplitude may be more important than call duration [29]."

The wording is somewhat confusing. First, that study found support for Zipf's law of abbreviation in males and the opposite trend in females. Second, a negative relationship between call amplitude and call usage is a form of law of abbreviation (there length or duration is replaced by a magnitude, amplitude in this case) and therefore is a prediction of compression based on the general mathematical theory in <https://arxiv.org/pdf/1906.01545.pdf>

References to research on hyraxes later on in the article need to be revised accordingly.

An important mathematical finding (<https://arxiv.org/pdf/1906.01545.pdf>) is that compression predicts that the correlation between frequency and magnitude cannot be (strictly positive) positive. Interestingly, optimal coding is possible even when the correlation is zero (non significant according to a two-sided correlation test). Then, although the positive correlation found in this article and previous research is not consistent with compression, compression cannot be excluded a priori for cases where no significant correlation is found (the two new world primates of Bezerra et al, ravens,...).

2. THE LAW OF ABBREVIATION

The authors investigate the relationship between phrase frequency and its duration arguing that phrases resemble human "words" more than notes.

For that reason, and for the sake of completeness and robustness, the authors should also investigate the relationship between phrase frequency and number of notes within the phrase, that is closer to the typical setup used for testing the law of abbreviation in human languages, where word length in letters or syllables is used. My prediction is that the analysis will confirm the current findings of an anti-law of abbreviation and thus the results will be more convincing for all sorts of language researchers.

3. REVIEW OF THE LITERATURE

The article lacks references to research on these laws in other species. For instance, concerning the law of abbreviation, the pioneering research by Hailman (about 30 years ago) on chick-a-dee's is missing. The recent findings by penguins (Favaro et al, Biolody letters) are also missing.

Such a problem can be fixed checking a comprehensive bibliography:

https://www.cs.upc.edu/~rferrericancholaws_of_language_outside_human_language.html.

line 81 "One of the first documented reports of Zipf's law of abbreviation in a nonhuman animal was in the repertoire of dolphin surface behaviors" needs to be revised accordingly.

In addition, the article would benefit from a wider perspective, other comparative research based on linguistic laws that are not the two laws considered by the authors. That would help to not lose historical perspective. For instance, the work by McCowan et al through Zipf's law for word frequencies (see biblio above).

line 92-94: Other well-known exceptions to the law are discussed here (fully body gestures in chimpanzees, heraldry,...)

<https://arxiv.org/pdf/1906.01545.pdf>

The case of the chimpanzees is particularly relevant for this article. It is discussed later on in the discussion section but could be cited quickly here.

4. HUMAN LANGUAGE UNIQUENESS

The statement "Human language is unique among communication systems as it contains semantics, wherein sounds are combined in unique ways to confer meaning [2]." is false in the way in its current formulation.

A possible counterexample (given the authors' loose statement): meaning in dolphins whistle sequences

- Ferrer-i-Cancho, R. & McCowan, B. (2009). A law of word meaning in dolphin whistle types. *Entropy* 11 (4), 688-701.

- Ferrer-i-Cancho, R. & McCowan, B. (2012). The span of correlations in dolphin whistle sequences. *Journal of Statistical Mechanics*, P06002.

Furthermore, could the authors state clearly the rigorous statistical test (traditional p-value testing? information theoretic model selection?...) that Berwick et al have used to determine that human language is unique in this regard? (I would like to apply it to my own data). If such a test did not exist the statement would be a projection of anthropocentric culture, not real science.

p. 8 "the diversity of language that is uniquely human [17]." Reference by Heesen et al is not about the diversity of language or the uniqueness of human language. It is rather the opposite, a challenge to anthropocentric biases and an attempt to provide unifying approaches.

The same applies to "the 'complexity and expressive power' of language is uniquely human [5]." I would really like to have such a statistical procedure.

In addition, indicating the actual range of species in which the test has been applied to determine the uniqueness of human language would really help. Have dolphins, humpback whales, belugas, ..., bats, geladas, ... been considered to reach such a conclusion?

It is also important to clarify, for a given species the modality or kind of vocalization (for instance, dolphins whistles versus click trains) that has been used to support the uniqueness claim.

Reviewer: 2

Comments to the Author(s)

This is an interesting paper investigating whether structural trade-offs known as Menzerath's and Zipf's laws apply to male gibbon songs. I believe that this study is a valuable addition to what we know about this topic. I only have few minor comments.

Abstract: Please provide the name of the study species (not just "gibbons").

L 35-48: I think that the opening paragraph, especially the second part of it that relates to human semantics, is not very relevant to what this paper is dealing with (e.g. this paper does not look at the relationship between call structure and meaning/function).

L 70-73: It is not clear how these compression-related constraints relate to syntax.

L 139-141: The last hypothesis does not seem to fit or follow the two previous ones related to trade-offs. I would suggest linking it to the previous two hypotheses better.

L 141-144: This is kind of repetition of what you said at the beginning of the Introduction.

Methods: Please start the Methods section with providing information on study subjects and study site, before describing data collection.

Figure 3: Please put full axis titles. Axis x title is missing.

L 308-309: While I understand that Spearman's rank correlation was used to be consistent with previous studies, this analysis is inappropriate due to, as you know, not dealing with the non-independence of datapoints, and therefore should not be reported (the fact that such analyses were provided by previous studies is not a justification for using them).

L 338: It is unclear to me what is meant by "coding efficiency" here, and what kind of selection pressures would act on it (ultimately leading to the described compression patterns). Maybe expand on this.

L 391-398. A substantial part of this is a repetition of what was said in the Introduction. Please try to make the text more concise by avoiding such redundancy.

L 399-409. It looks like the differences between the results of your and the other gibbon study are due to the differences in the call unit selected for analysis. But the title of the paper reads like your study shows a different pattern compared with that previous study. This does not need to be the case, since, as you say, you did a different kind of analysis. So the title framed the way you did might be a bit misleading.

L 409-421: As mentioned above, this part of the study seems to be disconnected from the rest. It is interesting but it reads like from a different paper. Maybe try to integrate this aspect better with the rest of the MS.

L 426-429: But maybe, as mentioned above, you would find this if you used a different unit for analysis.

L 437-438: Maybe just "different" rather than "much different". E.g. Menzerath's law seems to be supported by data from several animal taxa.

Author's Response to Decision Letter for (RSOS-200151.R0)

See Appendix A.

Decision letter (RSOS-200151.R1)

10-Mar-2020

Dear Professor Clink,

It is a pleasure to accept your manuscript entitled "Brevity is not a universal in animal communication: evidence for compression depends on the unit of analysis in small ape vocalizations" in its current form for publication in Royal Society Open Science.

on behalf of Dr Claudia Wascher (Associate Editor) and Kevin Padian (Subject Editor)
openscience@royalsociety.org

Appendix A

Reviewer comments to Author:

Reviewer: 1

Comments to the Author(s)

This is a very important contribution to research on the law of abbreviation and Menzerath's law in other species. This research builds directly on findings of these laws at the level of note usage and organization in crested gibbons. The authors extend the analysis to Bornean gibbons regarding phrases (made of notes) as a "words" and find that Menzerath's law holds while the law of abbreviation does not; actually, they find the inverse of the law of abbreviation (a positive correlation between frequency of use and phrase duration). The article is making a very important contribution towards understanding the conditions under which the principle of compression manifests. The article is of high technical quality, well-written and thus deserves publication. However, I have some particular concerns that I explain below (I think they can be addressed easily).

1. THEORETICAL BACKGROUND

The current reasoning is somewhat superficial from a theoretical standpoint.

Abstract: "Zipf's law of abbreviation (which predicts a negative relationship between signal length and frequency of use). Indeed, Zipf's law of abbreviation does not predict in a theoretical sense. What really predicts is the principle of compression. The law of abbreviation is a manifestation of compression. Making predictions based on the law of abbreviation per se is just inductive reasoning.

(but in lines 54-55 the authors are right when stating that "the principle of compression predicts ...")

We changed the wording in the abstract to be more precise"

'Zipf's law of abbreviation (wherein a negative relationship between signal length and frequency of use is expected)'

Abstract: "are linked to compression" -> "are predictions of compression".

We modified this line accordingly.

"We did not find support for Zipf's law of abbreviation—which predicts a negative relationship between duration of a signal and the frequency of use." It would be better to state that

"We did not find [evidence of compression for the] relationship between duration of a signal and the frequency of use."

Please, revise the use of "predict/ion..." everywhere.

We removed this line from the abstract for the sake of brevity, and were careful to revise our use of prediction elsewhere in the manuscript.

The title "Brevity is not a universal in animal communication: lack of evidence for efficient coding in phrases of small ape vocalizations"

Gibbons may be coding efficiently, but not to maximize compression but to maximize

transmission success, indicate fitness, ... (as reviewed by the authors in the discussion section).

A theoretically sounder title could be "Brevity is not a universal in animal communication: lack of evidence for [compression] in phrases of small ape vocalizations"

The problem of that title would be that Menzerath's law is still a prediction of compression (Gustison et al and <https://arxiv.org/pdf/1906.01545.pdf>).

Furthermore, the issue that the law of abbreviation is not universal in animal communication is not totally new. It failed in chimpanzees for body gestures, in female hyraxes (but no in males) and also failed in human heraldry.

Paradoxically, "brevity is not universal" (these gibbons do not show the law of abbreviation) but show another prediction of compression, i.e. Menzerath's law.

It is convenient to distinguish universality of the applicability of the principle of compression at two levels: (a) species level (b) all behaviors of a species (c) ...

I do not think that it is difficult to find violations of the predictions of compression in some dimension of the behavior of a species even in humans. What it will be really difficult is to show that no aspect of a species is driven by compression, even in experimental conditions or ecological contexts where compression should be favoured. That is the real point.

We agree with this very insightful point. We aimed to emphasize that evidence for compression appears to depend on the unit of analysis, but agree that it can depend on a multitude of factors, and that there will most likely be evidence for (or against) it across many species, contexts, behaviors. We modified our title accordingly (and also in line with the comment from reviewer #2):

'Brevity is not a universal in animal communication: evidence for compression depends on the unit of analysis in small ape vocalizations'

line 376-378 "It is possible that compression is not universally applicable in vocal communication systems, or that other selection pressures were stronger in shaping phrase organization within male solos."

A suitable citation here is Ferrer-i-Cancho et al in Cognitive Science (that is revised later on the article in detail).

We added the citation here.

line 386-388 "Although we found support for Menzerath's law, the underlying processes that led to the observed pattern are not clear. It is possible that the observed pattern is the result of selection for coding efficiency,"

Indeed, Gustison et al (cited) proposed a generalization of compression behind Menzerath's law and should be cited here.

We added the citation here.

The most updated version of the mathematical theory of the law abbreviation (and Menzerath's law) is found here

<https://arxiv.org/pdf/1906.01545.pdf>

Relevant, for instance, in line 458.

Thank you for this suggestion. We added this in addition to the Gustison et al paper.

lines 91-95 "A recent study on rock hyraxes did not find support Zipf's law of abbreviation, but rather found a negative relationship between call amplitude and call usage, and the authors posit that for long-distance communication costs of call amplitude may be more important than call duration [29]."

The wording is somewhat confusing. First, that study found support for Zipf's law of abbreviation in males and the opposite trend in females. Second, a negative relationship between call amplitude and call usage is a form of law of abbreviation (there length or duration is replaced by a magnitude, amplitude in this case) and therefore is a prediction of compression based on the general mathematical theory in <https://arxiv.org/pdf/1906.01545.pdf>

We modified this line to the following based on the reviewer input:

'that these findings are consistent with a form of Zipf's law of abbreviation (wherein length or duration is replaced amplitude) and is therefore consistent with predictions of compression (Demartsev et al., 2019)'.

References to research on hyraxes later on in the article need to be revised accordingly. Thank you for catching this. We clarified the description of rock hyrax work throughout.

An important mathematical finding (<https://arxiv.org/pdf/1906.01545.pdf>) is that compression predicts that the correlation between frequency and magnitude cannot be (strictly positive) positive. Interestingly, optimal coding is possible even when the correlation is zero (non significant according to a two-sided correlation test). Then, although the positive correlation found in this article and previous research is not consistent with compression, compression cannot be excluded a priori for cases where no significant correlation is found (the two new world primates of Bezerra et al, ravens,...).

As mentioned above we added a citation for this.

2. THE LAW OF ABBREVIATION

The authors investigate the relationship between phrase frequency and its duration arguing that phrases resemble human "words" more than notes.

For that reason, and for the sake of completeness and robustness, the authors should also investigate the relationship between phrase frequency and number of notes within the phrase, that is closer to the typical setup used for testing the law of abbreviation in human languages, where word length in letters or syllables is used. My prediction is that the analysis will confirm the current findings of an anti-law of abbreviation and thus the results will be more convincing for all sorts of language researchers.

We included this analysis along with our earlier analysis of Zipf's law of abbreviation. As the reviewer predicted, we did not find a negative relationship between phrase use and

number of notes in the phrase. But, in contrast to the analysis with mean note duration as the outcome, we did not find any relationship (positive or negative) as our null model was the top ranked model. We added a second panel to Figure 6 and updated our methods and results.

3. REVIEW OF THE LITERATURE

The article lacks references to research on these laws in other species. For instance, concerning the law of abbreviation, the pioneering research by Hailman (about 30 years ago) on chick-a-dee's is missing. The recent findings by penguins (Favaro et al, Biolody letters) are also missing.

Such a problem can be fixed checking a comprehensive bibliography: https://www.cs.upc.edu/~rferrericancholaws_of_language_outside_human_language.html.

line 81 "One of the first documented reports of Zipf's law of abbreviation in a nonhuman animal was in the repertoire of dolphin surface behaviors" needs to be revised accordingly.

We thank the reviewer for bringing this to our attention, and we have modified the text as outlined below:

'One of the first documented tests of Zipf's law of abbreviation in a nonhuman animal was in black-capped chickadees, and in this case adherence to the law was dependent on the level of analysis: use of different call types was not negatively correlated with the number of notes, but bouts (which are composed of multiple calls with at least a 30-s break in between subsequent calls) did follow Zipf's law of abbreviation (Hailman et al., 1985).

Zipf's law of abbreviation was subsequently shown in the repertoire of dolphin surface behaviors (Ferrer-i-Cancho and Lusseau, 2009), vocalizations of Formosan macaques (Semple et al., 2010), short-range bat vocalizations (Luo et al., 2013), in subsets of the gestural repertoire of chimpanzees (Heesen et al., 2019) and penguins (Favaro et al., 2020)'.

In addition, the article would benefit from a wider perspective, other comparative research based on linguistic laws that are not the two laws considered by the authors. That would help to not lose historical perspective. For instance, the work by McCowan et al through Zipf's law for word frequencies (see biblio above).

Thank you for this suggestion. We have slightly reworked the introduction so that there is less emphasis on semantics (as suggested by reviewer #2) and added a section related to Zipf's law.

line 92-94: Other well-known exceptions to the law are discussed here (fully body gestures in chimpanzees, heraldry,...)

<https://arxiv.org/pdf/1906.01545.pdf>

The case of the chimpanzees is particularly relevant for this article. It is discussed later on in the discussion section but could be cited quickly here.

We added a line about chimpanzee gestures here.

4. HUMAN LANGUAGE UNIQUENESS

The statement "Human language is unique among communication systems as it contains semantics, wherein sounds are combined in unique ways to confer meaning [2]." is false in the way in its current formulation.

A possible counterexample (given the authors' loose statement): meaning in dolphins whistle sequences

- Ferrer-i-Cancho, R. & McCowan, B. (2009). A law of word meaning in dolphin whistle types. *Entropy* 11 (4), 688-701.
- Ferrer-i-Cancho, R. & McCowan, B. (2012). The span of correlations in dolphin whistle sequences. *Journal of Statistical Mechanics*, P06002.

Furthermore, could the authors state clearly the rigorous statistical test (traditional p-value testing? information theoretic model selection?...) that Berwick et al have used to determine that human language is unique in this regard? (I would like to apply it to my own data). If such a test did not exist the statement would be a projection of anthropocentric culture, not real science.

We appreciate the reviewer's clarification of this matter. We have modified the lines and included the relevant citations them:

'Human language has been considered unique among communication systems as it contains semantics, wherein sounds are combined in unique ways to confer meaning (Hurford, 2011), but there have been cases wherein vocalizations have been linked to particular behavioral contexts (Ferrer-i-Cancho and McCowan, 2009), and the topic of semantics in animal vocalizations remains a topic of debate (Manser, 2016)'.

p. 8 "the diversity of language that is uniquely human [17]." Reference by Heesen et al is not about the diversity of language or the uniqueness of human language. It is rather the opposite, a challenge to anthropocentric biases and an attempt to provide unifying approaches.

We changed the citation to the following which does outline ways in which human language is unique among animals.

Hurford, J.R., 2011. *The origins of grammar: Language in the light of evolution II*. OUP Oxford.

The same applies to "the 'complexity and expressive power' of language is uniquely human [5]." I would really like to have such a statistical procedure.

We changed the language slightly so that it is less certain and indicates that it is something that has been proposed:

'but it has been proposed that the 'complexity and expressive power' of language is uniquely human'

In addition, indicating the actual range of species in which the test has been applied to determine the uniqueness of human language would really help. Have dolphins, humpback whales, belugas, ..., bats, geladas, ... been considered to reach such a conclusion?

It is also important to clarify, for a given species the modality or kind of vocalization (for instance, dolphins whistles versus click trains) that has been used to support the uniqueness claim.

We completely agree, and in fact had similar thoughts in the writing of this manuscript. Although addressing this is outside of the scope of the present manuscript, we worked hard to tone down the language we used when attributing something as being uniquely human.

Reviewer: 2

Comments to the Author(s)

This is an interesting paper investigating whether structural trade-offs known as Menzerath's and Zipf's laws apply to male gibbon songs. I believe that this study is a valuable addition to what we know about this topic. I only have few minor comments.

Abstract: Please provide the name of the study species (not just "gibbons").

We added this to the abstract.

L 35-48: I think that the opening paragraph, especially the second part of it that relates to human semantics, is not very relevant to what this paper is dealing with (e.g. this paper does not look at the relationship between call structure and meaning/function).

We modified the introductory paragraph to shift away from a focus on semantics to more of a comparison between human/animal models.

L 70-73: It is not clear how these compression-related constraints relate to syntax.

We agree that this was not clear as written, and have modified the lines

'The evolutionary origin of human language is not well understood (Sainburg et al., 2019), and a better understanding of the way in which animals combine singular sounds into complex sequences, along with the tradeoffs that constrain vocal production (Fedurek et al., 2017), has important implications for understanding variation in acoustic complexity across taxa (Gustison et al., 2016; Kershenbaum et al., 2016)'.

L 139-141: The last hypothesis does not seem to fit or follow the two previous ones related to trade-offs. I would suggest linking it to the previous two hypotheses better.

We slightly modified the lines to better integrate this section.

And lastly, we investigated the degree of inter-individual variation in phrase types, given previously documented individual-signatures in note type use (Geissmann et al., 2005; Inoue et al., 2017; Mitani and Marler, 1989) and phrases (Fan et al., 2011; Wanelik et al., 2012).

L 141-144: This is kind of repetition of what you said at the beginning of the Introduction.

We modified the line accordingly:

'The continued exploration of the applicability of statistical laws developed for human language in nonhuman systems, particularly in closely related species such as gibbons, can provide insights regarding the evolutionary history of universal linguistic patterns and the evolutionary precursors that lead to the diversity of human languages (Hurford, 2011)'.

Methods: Please start the Methods section with providing information on study subjects and study site, before describing data collection.

We have added a section describing the study subjects and have included a map of recording locations.

Figure 3: Please put full axis titles. Axis x title is missing.

Thank you for catching this. We modified the figure and it now has the x-axis label.

L 308-309: While I understand that Spearman's rank correlation was used to be consistent with previous studies, this analysis is inappropriate due to, as you know, not dealing with the non-independence of datapoints, and therefore should not be reported (the fact that such analyses were provided by previous studies is not a justification for using them).

We agree and have removed this analysis from the manuscript.

L 338: It is unclear to me what is meant by "coding efficiency" here, and what kind of selection pressures would act on it (ultimately leading to the described compression patterns). Maybe expand on this.

We removed this line for clarity.

L 391-398. A substantial part of this is a repetition of what was said in the Introduction. Please try to make the text more concise by avoiding such redundancy.

We agree and removed the species-specific discussion and outlined more broadly cases in which there were patterns which were consistent and inconsistent with Zipf's law.

L 399-409. It looks like the differences between the results of your and the other gibbon study are due to the differences in the call unit selected for analysis. But the title of the paper reads like your study shows a different pattern compared with that previous study. This does not need to be the case, since, as you say, you did a different kind of analysis. So the title framed the way you did might be a bit misleading.

After careful deliberation we decided to change the title based on both reviewer comments. The title is now as follows:

'Brevity is not a universal in animal communication: evidence for compression depends on the unit of analysis in small ape vocalizations'

L 409-421: As mentioned above, this part of the study seems to be disconnected from the rest. It is interesting but it reads like from a different paper. Maybe try to integrate this aspect better with the rest of the MS.

We added this line in an attempt to better link this section to the others:

‘The evolutionary implications of adherence (or lack thereof) to Zipf’s law of abbreviation are unclear, and in gibbons it is possible that selection pressure for phrases of longer duration have been stronger than the selection pressure for compression’.

L 426-429: But maybe, as mentioned above, you would find this if you used a different unit for analysis.

As mentioned above we tried to emphasize the importance of the unit of analysis, but chose to leave this line as-is.

L 437-438: Maybe just "different" rather than "much different". E.g. Menzerath's law seems to be supported by data from several animal taxa.

We modified this line accordingly.